# Time-series analysis for forecasting monthly workload at two elephant hospitals in Thailand

**Worapong Kosaruk**[1,2]*, **Veerasak Punyapornwithaya**[1], **Chatchote Thitaram**[1,2], **Pichamon Ueangpaibool**[3], **Nawapa Hirannithithamrong**[4]

1 Faculty of Veterinary Medicine, Chiang Mai University, Chiang Mai, Thailand, 2 Elephant, Wildlife and Companion Animals Research Group, Chiang Mai University, Chiang Mai, Thailand, 3 National Elephant Institute, Elephant Hospital, Thai Elephant Conservation Center, Forest Industry Organization, Ministry of Natural Resources and Environment, Lampang, Thailand, 4 National Institute of Elephant Research and Health Service, Division of Animal Welfare and Veterinary Service, Department of Livestock Development, Surin, Thailand

* worapong.kosa@cmu.ac.th

## Abstract

The increasing demand for elephant healthcare in Thailand underscores the need for efficient resource management in veterinary hospitals. Treating sick elephants requires substantial medical inputs, and without accurate workload forecasting, hospitals risk shortages in staff, equipment, and medications, potentially compromising care quality. This study evaluated six time-series forecasting models— Autoregressive Integrated Moving Average and its seasonal model (ARIMA/SARIMA), Exponential Smoothing (ETS), Trigonometric, Box-Cox ARMA Trend Seasonality (TBATS), Neural Network Time Series Regression (NNTR), Prophet, and Extreme Gradient Boosting (XGBoost)—to predict monthly workloads at two major elephant hospitals: the National Elephant Institute (NEI) hospital in northern Thailand and the Department of Livestock Development (DLD) hospital in the northeast. Historical admission data spanning five years (NEI) and nine years (DLD) were analyzed. The ETS model achieved the highest accuracy at NEI, outperforming all other models by effectively capturing its stable and seasonal caseload patterns. In contrast, the NNTR model performed best at DLD, where it accommodated irregular fluctuations likely driven by external factors. Forecasts for 2025 suggested a consistent caseload at NEI (28–31 cases/month) and a declining trend at DLD, with the lowest projection in September. Although patient volume was low, each case demanded disproportionately high resources, justifying the need for anticipatory planning. This study provides the first demonstration that multi-model time-series forecasting can generate actionable insights using sparse, real-world data from wildlife hospitals. By embedding predictive analytics into routine operations, it offers a novel, scalable framework to support data-driven decision-making in endangered species care.

**Data availability statement:** The raw admission data used in this study cannot be publicly deposited due to privacy and regulatory restrictions. Access to these restricted data can be requested through a non-author institutional contact: Research Administration, Academic Service, and International Relations Section Faculty of Veterinary Medicine, Chiang Mai University Email: saraban_vet@cmu.ac.th The analytical R code and forecasting workflow are openly available at 10.5281/zenodo.16734811.

**Funding:** This study was partially supported by Chiang Mai University. The funders had no role in study design, data collection and analysis, decision to publish, or preparation of the manuscript. No additional external funding was received for this study.

**Competing interests:** The authors have declared that no competing interests exist.

## 1. Introduction

The Asian elephant (*Elephas maximus*) holds profound cultural and ecological importance in Thailand and is classified as Endangered by the IUCN [1,2]. Within the country, two distinct populations exist: a wild population of approximately 3,126−3,341 individuals inhabiting fragmented forests with ongoing human-elephant conflict, and a captive population of about 3,783 elephants integrated into human-managed society [2,3]. Captive elephants are central to Thailand's wildlife-based tourism industry, attracting millions of international visitors and contributing substantial economic values [1,4]. To safeguard their health and welfare, the Thai government has established specialized elephant hospitals that serve as critical infrastructure for veterinary care [1,5].

The National Elephant Institute (NEI) hospital, founded in 1993 by the Forest Industry Organization (FIO), remains the primary referral center for elephants in the northern Thailand [5]. NEI provides both mobile and admission-based care to an estimated captive population of 1,278 elephants [5,6]. In northeastern Thailand, the Department of Livestock Development (DLD) operates another major elephant hospital in Surin province, home to approximately 722 captive elephants (Unpublished data, Department of Provincial Administration of Thailand, 2024). This population is deeply rooted in regional cultural traditions, with local communities historically engaged in elephant handling [7]. Like NEI, the DLD hospital provides both mobile and inpatient services, primarily for complex or critical cases. These two hospitals collectively manage 20–100 new admissions per year, with monthly peaks reaching up to 30 cases. During periods of high demand, they frequently operate near full capacity, underscoring the need for robust caseload forecasting to optimize staffing, treatment preparation, and resource allocation.

Providing medical care for elephants requires substantial medical resources. For instance, daily fluid maintenance for an adult elephant may reach 200 liters [8], compared to just 2–3 liters for an adult large-breed dog [9]. Disease conditions such as Elephant Endotheliotropic Herpesvirus Hemorrhagic Disease (EEHV-HD), a fatal infectious disease in young calves, exhibited seasonal peaks (June-July) in Thailand and place disproportionate strain on hospital resources [10–12]. Likewise, gastrointestinal conditions especially colic, are linked to seasonal roughage quality, suggesting that environmental factors may play a significant role in elephant health [13,14]. These recurring, resource-intensive cases demand forward-looking systems to support continuous, high-quality care.

Despite these challenges, elephant hospitals in Thailand currently operate under reactive case-by-case models with limited use of predictive tools. This constrains their ability to anticipate seasonal fluctuations, plan for critical case surges, and maintain consistent medical provisioning. To address this gap, time-series forecasting offers a solution framework to predict hospital workload trends, optimize hospital operations and resource distribution [15–19]. In human health systems, time-series forecasting is widely used to anticipate hospital admissions and optimize healthcare delivery [16,17,20–23]. However, its use in veterinary medicine remains limited [24–26], and applications in zoo and wildlife hospitals are largely unexplored.

Although elephant hospitals serve fewer patients than human facilities, the high per-patient cost and the species' conservation significance make proactive forecasting no less critical. In addition, proactive forecasting is particularly relevant for recurring, high-cost conditions such as EEHV-HD and colic, commonly seen in Thai captive elephants, which often drive intense medical resource requirements.

This study applies six time-series models to predict monthly workloads at two major elephant hospitals in Thailand (NEI and DLD). Historical admission records were analyzed to identify workload patterns and compare model performance across two settings with distinct caseload structures. The models include Autoregressive Integrated Moving Average (ARIMA) and Seasonal ARIMA (SARIMA) [17,27], Exponential Smoothing (ETS) [20,21], Trigonometric, Box-Cox transformation, ARMA errors, Trend, and Seasonal components (TBATS) [20], Neural Network Time Series Regression (NNTR) [28], Prophet [29], and Extreme Gradient Boosting (XGBoost) [30]. While elephant hospital datasets are inherently small due to the rarity of hospitalizations, this study demonstrates that robust forecasting remains feasible and informative. To our knowledge, this is the first study to systematically evaluate and compare multiple forecasting models using sparse, real-world data from wildlife hospitals, offering a novel tool for strategic care planning. By integrating predictive analytics into elephant hospital workflows, this approach could enable a shift from reactive care to proactive planning, supporting resource allocation, staffing, and preparedness in endangered species healthcare.

## 2. Materials and methods

### 2.1. Study sites and data sources

This study analyzed historical elephant admission data from two major government-operated elephant hospitals in Thailand: 1) NEI hospital in northern Thailand (managed by FIO; coordinates 18°21'25.2"N 99°14'52.1"E) and 2) DLD hospital in northeastern Thailand (managed by the National Institute of Elephant Research and Health Service; coordinates 14°46'28.8"N 103°26'09.3"E). NEI hospital, located in a forested area, can accommodate up to 20 elephants at a time. Its spacious, nature-integrated layout allows stable cases to be housed beyond the core hospital unit when needed. In contrast, DLD hospital is situated in a constrained urban area and can manage no more than three elephants concurrently.

Historical data availability constrained our dataset selection. Hospital records spanning five and nine consecutive years were analyzed for NEI (year 2020–2024, N = 467) and DLD (year 2016–2024, N = 198), respectively. These study periods were chosen because record-keeping prior to 2016 (DLD) and 2020 (NEI) was predominantly paper-based and inconsistent; thus, we focused on the years with the most complete and reliable data for analysis.

Individual case data included admission and discharge dates, diagnosis, demographics (age, sex, and camp origin), and treatment outcomes. Preventive care visits and cases with incomplete records were excluded to ensure data consistency. Data was analyzed separately in two elephant hospitals. Monthly "active cases" were computed based on admission-discharge intervals, proportionally allocated across months to reflect real-time workload. All data were anonymized prior to analysis. Data processing involved digitization, verification, and categorical coding. All analyses and modeling were conducted in R (version 4.0.0).

### 2.2. Forecasting workflow

The time-series forecasting workflow (Fig 1) consisted of four key steps: 1) decomposition of time-series data into trend and seasonal components, 2) development of forecasting models, 3) evaluation and selection of the best-performing model based on accuracy metrics, and 4) forecasting the monthly hospital workload for January to December 2025.

### 2.3. Time-series decomposition and seasonality testing

Descriptive statistic was calculated using the dplyr package [31], reported as mean ± standard error of mean (SE). For model development, each hospital dataset was divided into training set (NEI: 2020–2023; DLD: 2016–2023) and a test set (January to December 2024). To explore underlying temporal patterns, an additive decomposition of the monthly workload

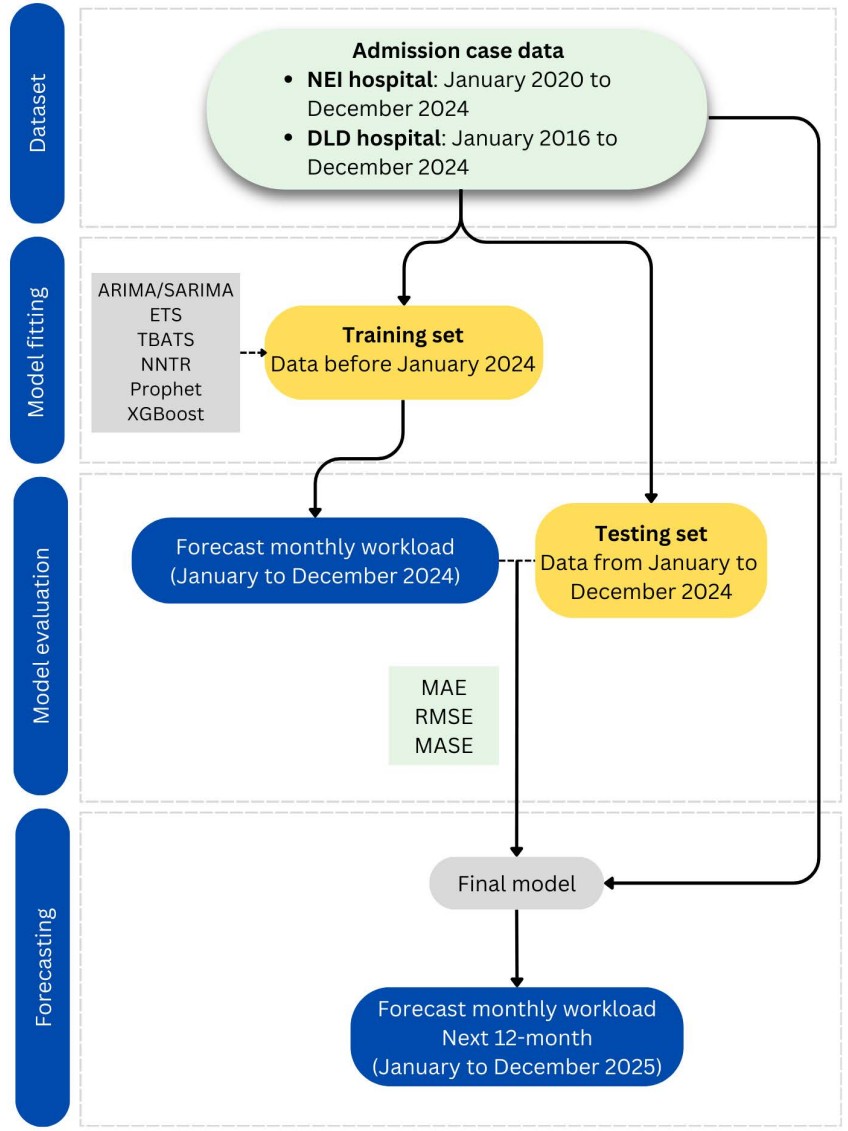

**Fig 1. Workflow of time-series modeling.** ARIMA = Auto Regressive Integrated Moving Average model, SARIMA = Seasonal ARIMA model, ETS = Exponential Smoothing model, TBATS = Trigonometric, Box-Cox transformation, ARMA errors, Trend, and Seasonal components model, NNTR = Neural Network Time Series Regression, Prophet = Prophet model, XGBoost = Extreme Gradient Boosting, MAE = Mean Absolute Error, RMSE = Root Mean Square Error, MASE = Mean Absolute Scaled Error.

was performed using the stats package in R [32]. Each time-series was separated into trend, seasonal, and residual components. Formal assessment of seasonality was conducted using Ollech and Webel's combined seasonality test (WO-test) from the seastests package [33], with a significance threshold of P < 0.01.

To further characterize the underlying structure of each dataset, statistical comparisons between NEI and DLD hospital data were conducted. Descriptive comparisons included age, length of stay (LOS), and active monthly caseloads. Welch's *t*-test and Wilcoxon rank-sum test were used for continuous variables (e.g., age, LOS), while variance differences in monthly caseloads were assessed using Levene's test. Additionally, autocorrelation patterns were explored using autocorrelation function (ACF) plots to assess temporal dependencies.

## 2.4. Model descriptions

Six time-series models were employed to predict the monthly hospital workload at both study sites. These forecasting models were chosen to cover diverse analytical approaches. Classical statistical methods (ARIMA/SARIMA, ETS, TBATS) were selected to handle linear and seasonal patterns, while machine learning-based methods (NNTR, Prophet, XGBoost) were included to capture potential non-linear relationships and more complex data structures.

**ARIMA and SARIMA**: The Autoregressive Integrated Moving Average (ARIMA) or the Seasonal ARIMA (SARIMA) models [22] were utilized for modeling both trend and seasonal patterns. To ensure the appropriate model selection, the WO-test was conducted prior to analysis. If significant, the SARIMA model was used; otherwise, the ARIMA model was applied.

The ARIMA model is defined as:

$$(1 - \phi_1 L - \phi_1 L^2 - \ldots - \phi_p L^p)(1 - L)^d Y_t = (1 + \theta_1 L + \theta_2 L^2 + \ldots + \theta_q L^q) \in_t$$

where $\phi$ represents autoregressive coefficients, $\theta$ represents moving average coefficients, $d$ denotes the differencing order, $\in_t$ is the error term at time $t$ [34].

SARIMA model extends ARIMA by incorporating seasonal components. SARIMA model can be defined as: $SARIMA(p, d, q)(P, D, Q)_s$, where $P$, $D$, and $Q$ represent the seasonal autoregressive, differencing, and moving average terms, respectively, and $s$ denotes the seasonal period (e.g., 12 for monthly data) [35]. These models were implemented using the auto.arima() function from the focastHybrid package in R [36].

**Exponential Smoothing (ETS) model** [23]: the ETS model estimates trends and seasonal patterns by applying weighted averages that assign more importance to recent observations, making it effective for datasets with stable or slowly evolving patterns. The ETS model can be expressed as: $Y_t = E_t + T_t + S_t$, where $E_t$, $T_t$, and $S_t$ represent the error, trend, and seasonality components, respectively, which can be combined additively or multiplicatively based on the data characteristics [37]. ETS was implemented using the ets() function in R [36].

**Trigonometric, Box-Cox transformation, ARMA errors, Trend, and Seasonal components (TBATS) model** [20]: TBATS accommodates complex seasonal patterns, including those with multiple seasonality or irregular trends. This model decomposes the time series into trigonometric representations of seasonality, applies a Box-Cox transformation for variance stabilization, and models residual errors using autoregressive moving average (ARMA) processes. The model expressed as:

$$Y_t^{(\omega)} = Trend_t + Seasonal\ Term_t + ARMA\ Error_t$$

where $Y_t^{(\omega)}$ is the observed value at time $t$ after applying the Box-Cox transformation with parameter $\omega$, which stabilizes variance and ensures linearity, $Trend_t$ represents the trend component, $Seasonal\ Term_t$ consists of seasonal components modeled using trigonometric functions, and $ARMA\ Error_t$ captures any remaining errors after accounting for the trend and seasonality using ARMA structure [38]. The model was fitted using the tbats() function from the forecast package in R [36].

**Neural Network Time Series Regression (NNTR) model** [20,39]: This model used a feed-forward neural network with one hidden layer, selecting lagged input variables based on the partial autocorrelation function (PACF) to represent key temporal dependencies [40]. Although NNTR lacks a mathematical formula, its architecture can be represented algorithmically as:

$$Y_t = f_L \left( f_{L-1} \left( \ldots f_1 \left( Y_{t-1}, Y_{t-2}, \ldots, Y_{t-p}; W_1 \right); W_2 \right) \ldots; W_L \right) + \in_t$$

where $f_1, f_2, \ldots, f_L$ represent the activation functions of each layer, $W_1, W_2, \ldots, W_L$ are the learned weights, $Y_{t-1}, Y_{t-2}, \ldots, Y_{t-p}$ are the input features, and $\in_t$ is the error term. The NNTR model was implemented using the nnetar() function in R [36]. Bootstrapping was applied to generate the 95% confidence interval for the model predictions.

**Prophet** [29]: This model decomposes the time series into three components: trend ($g(t)$), seasonality ($s(t)$), and holidays or events ($h(t)$), represented as $Y_t = g(t) + s(t) + h(t) + \in_t$, where $\in_t$ denotes the error term. While designed for

datasets with seasonality, Prophet's flexibility in trend modeling allows it to perform well even in datasets with no evident seasonal patterns [41]. The model was implemented using the prophet package in R [42], with default priors adjusted to match the characteristics of the data.

**Extreme Gradient Boosting (XGBoost)** [30]: This machine learning algorithm was applied to capture complex, non-linear relationships in the data. It uses an ensemble of decision trees, with each tree corrects the errors of the previous one. The final prediction is expressed as:

$$\hat{Y}_t = \sum_{k=1}^{K} f_k(X_t)$$

where $f_k$ represents the $k$-th tree, and $X_t$ are the input variables. XGBoost excels in handling non-linear trends, interactions among variables, and datasets with zero-inflated or irregular counts. In this study, time-lagged features were generated to capture temporal dependencies, and the optimal parameters of the XGBoost model were determined through grid search. Bootstrapping was also applied to estimate the 95% confidence interval. The implementation was performed using the xgboost package in R [43].

## 2.5. Model evaluation and forecasting

All models were trained on pre-2024 data (NEI: 2020–2023; DLD: 2016–2023) and tested on 2024 data. Forecast accuracy was evaluated using three metrics: Mean Absolute Error (MAE), Root Mean Squared Error (RMSE) and Mean Absolute Scaled Error (MASE). MAE measures simple average errors; RMSE penalizes larger errors more significantly; MASE provides a scale-independent error measure, comparing model forecasts against a naïve benchmark. The best-performing model for each hospital (lowest error matrices) was retrained on the full dataset and used to forecast monthly workload for January-December 2025.

## 3. Results

### 3.1. Descriptive and time-series components

From January 2020 and December 2024, a total of 467 elephant cases were admitted to NEI elephant hospital. Of these, 258 (55.25%) were males (mean age: 32.92 ± 1.62 years, range: 1 day-68 years), and 209 (44.75%) were females (mean age: 34.25 ± 1.25 years, range: 3 months-68 years). Admission primarily included integumentary issues (N = 131, 28.05%), gastrointestinal problems (N = 119, 25.48%), ocular disorders (N = 65, 13.92%), and musculoskeletal conditions (N = 47, 10.06%). Less frequent conditions were elephant endotheliotropic herpesvirus hemorrhagic disease (EEHV-HD, N = 19, 4.06%), reproductive issues (N = 16, 3.43%), toxicity (N = 15, 3.21%), dental/oral problems (N = 10, 2.14%), and urinary/renal disorders (N = 6, 1.28%), respiratory and neurological conditions (each N = 3, 0.64%), and metabolic/endocrine disorders (N = 2, 0.43%). Additionally, miscellaneous conditions such as weakness and inappetence accounted for 31 admissions (6.63%). The monthly workload at NEI hospital averaged 30.43 ± 0.10 cases (range: 28–31 active cases per month).

The DLD hospital admitted 198 individual elephants between January 2016 and December 2024, including 81 males (40.91%, mean age: 19.20 ± 2.09 years, range: 4 days-76 years) and 117 females (59.09%, mean age: 24.94 ± 1.69 years, range: 1–80 years). The leading cause of admission was gastrointestinal issues (N = 80, 40.40%), followed by integumentary problems (N = 43, 21.72%), musculoskeletal disorders (N = 20, 10.10%), and ocular conditions (N = 11, 5.55%). Other medical issues included EEHV-HD (N = 7, 3.53%), toxicity (N = 6, 3.03%) and miscellaneous conditions (N = 14, 7.07%). Additionally, 17 elephants (8.64%) were clinically healthy, accompanying sick individuals and utilizing hospital resources. The monthly workload at DLD hospital averaged 18.68 ± 1.13 cases (range: 0–31 cases per month).

Descriptive and statistical comparisons between hospitals confirmed structural differences between datasets (S1 Table). Elephants at NEI were significantly older and had shorter but more regular stays, while DLD exhibited highly variable monthly caseloads and minimal autocorrelation (S1 Fig), reflecting stochastic admission patterns.

The decomposition of the monthly workload at NEI and DLD elephant hospitals is shown in Fig 2. NEI elephant hospital showed a relatively stable workload pattern, with minor fluctuations and consistent seasonal peaks occurring regularly throughout the study period. Trend analysis indicated relatively stable workload levels, while the seasonal component showed a consistent annual cycle confirmed by significant WO-test results (P < 0.01). Conversely, DLD hospital workloads displayed greater variability, with fluctuating trends and no statistically significant seasonality (WO-test, P > 0.01).

### 3.2. Model performance

The performance of six forecasting models (ARIMA/SARIMA, ETS, TBATS, NNTR, Prophet, XGBoost) was evaluated using error metrics on training and testing datasets for each hospital (Table 1; Fig 3 and Fig 4). The ETS model consistently achieved the lowest forecasting errors (MAE, RMSE, MASE) at NEI hospital, demonstrating superior linear and seasonal forecasting capability. Thus, ETS was selected as the final model for NEI hospital. For DLD elephant hospital, the NNTR model provided superior predictive performance, effectively capturing non-linear patterns and demonstrating strong generalizability during testing. Consequently, NNTR was chosen as the optimal forecasting model for DLD hospital.

### 3.3. Workload forecasting

The ETS model forecasted NEI hospital's monthly workload for January-December 2025 (Fig 5). Predicted monthly admissions averaged 30.44 ± 0.23 cases, with the highest workload in January (31.07 cases), and lowest in February (28.39 cases). Monthly admissions thereafter remained stable, ranging from 29.96 to 31.02 cases.

For DLD hospital, the NNTR model forecasted an average monthly workload of 13.71 ± 0.47 cases in 2025 (Fig 6). The forecast showed a peak in January (18.27 cases), a sharp decline in February (14.49 cases), followed by a gradual decrease reaching its lowest point in September (12.04 cases). Admissions rose slightly again in October (12.83 cases), peaked in November (13.87 cases), and decreased in December (12.59 cases).

## 4. Discussion

This study presents a novel application of predictive analytics in captive Asian elephant healthcare, demonstrating that it is both feasible and informative to forecast monthly hospital admissions using sparse, real-world datasets. By systematically evaluating six time-series models across two elephant hospitals in Thailand—NEI (north) and DLD (northeast)— which exhibited contrasting patterns in disease prevalence, seasonality, and caseload dynamics. NEI showed consistent seasonal trends, best modeled by ETS, while DLD displayed irregular fluctuations, more accurately captured by NNTR. These differences underscore the importance of tailoring forecasting approaches to the specific data structure of each setting. Importantly, this study provides one of the first demonstrations that predictive forecasting frameworks, common in human health systems, can be effectively adapted for wildlife hospitals that manage few patients but require disproportionately high resource use per case. By demonstrating that robust, model-driven forecasts are possible even with limited datasets, and can directly inform hospital-level decision-making in conservation settings. This approach offers tangible value in optimizing staffing, supply chains, and emergency preparedness in endangered species care.

### 4.1. Hospital-specific caseload patterns

NEI's stable workload reflects its role as both a referral center and long-term care facility. The hospital manages approximately 125 elephants, most of which are geriatric and previously employed in the logging industry before transitioning into tourism roles [1,4]. This results in a high prevalence of chronic degenerative conditions, including osteoarthritis,

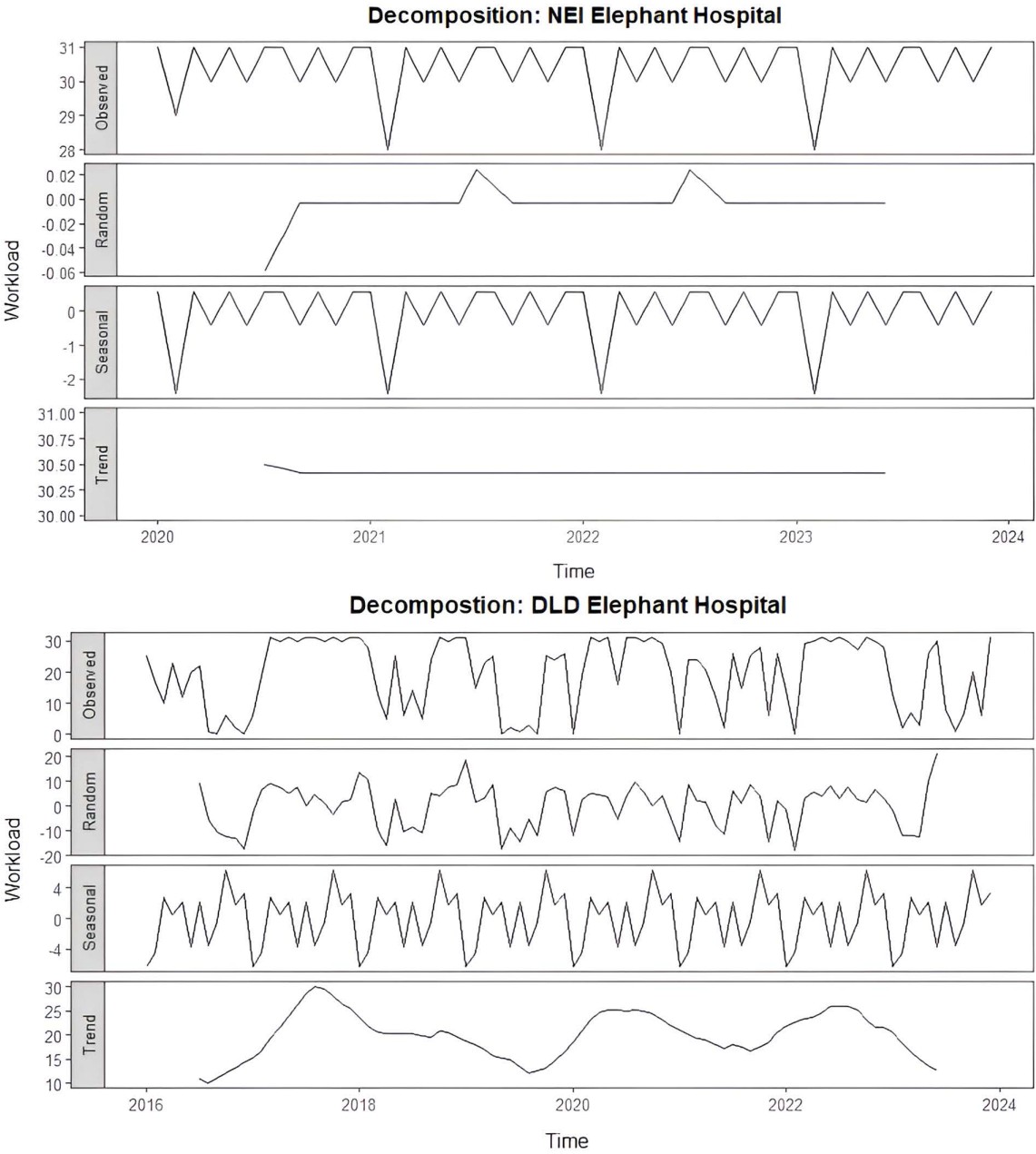

**Fig 2. Decomposition of monthly workload data at NEI (2020-2024) and DLD (2016-2024) elephant hospitals into observed, random, seasonal, and trend components.**

weakness, or persistent wounds, requiring prolonged treatment and continuous medical oversight. In December 2024, 23 elephants were under treatment at NEI, of which 13 were from the FIO, six from the Department of National Parks (DNP), and one from the Royal Elephant Stable (RES), and only three from external tourist facilities. These institutional affiliations contribute to NEI's consistent caseload and align with the superior performance of ETS, which is well-suited to structed, stable datasets.

**Table 1. Error metrics for time series models applied to the training data (January 2020 to December 2023 for NEI elephant hospital and January 2016 to December 2023 for DLD elephant hospital) to forecast monthly workload for January to December 2024, compared with actual values from the test dataset (January to December 2024).**

| Model | NEI elephant hospital | | | | | | DLD elephant hospital | | | | | |
|---|---|---|---|---|---|---|---|---|---|---|---|---|
| | Training data (January 2020 to December 2023) | | | Testing data (January to December 2024) | | | Training data (January 2016 to December 2023) | | | Testing data (January to December 2024) | | |
| | MAE | RMSE | MASE | MAE | RMSE | MASE | MAE | RMSE | MASE | MAE | RMSE | MASE |
| ARIMA/SARIMA | 0.028 | 0.145 | 0.025 | 0.167 | 0.408 | 0.167 | 8.704 | 9.968 | 1.039 | 12.080 | 13.714 | 1.219 |
| ETS | 0.061 | 0.127 | 0.053 | 0.160 | 0.368 | 0.160 | 8.403 | 10.750 | 1.003 | 13.160 | 14.738 | 1.328 |
| TBATS | 0.067 | 0.122 | 0.058 | 0.217 | 0.400 | 0.217 | 8.425 | 10.758 | 1.006 | 13.132 | 14.694 | 1.325 |
| NNTR | 0.028 | 0.166 | 0.024 | 0.167 | 0.409 | 0.167 | 11.934 | 14.188 | 1.424 | 9.762 | 11.631 | 0.985 |
| Prophet | 0.041 | 0.073 | 0.159 | 0.183 | 0.351 | 0.183 | 9.056 | 10.540 | 1.682 | 14.097 | 15.486 | 1.423 |
| XGBoost | 0.308 | 0.466 | 0.276 | 0.368 | 0.518 | 0.368 | 0.538 | 1.154 | 0.064 | 15.387 | 16.620 | 1.553 |

ARIMA = Auto Regressive Integrated Moving Average model, SARIMA = Seasonal ARIMA model, ETS = Exponential Smoothing model, TBATS = Trigonometric, Box-Cox transformation, ARMA errors, Trend, and Seasonal components model, NNTR = Neural Network Time Series Regression, Prophet = Prophet model, XGBoost = Extreme Gradient Boosting, MAE = Mean Absolute Error, RMSE = Root Mean Square Error, MASE = Mean Absolute Scaled Error.

In contrast, DLD hospital does not maintain a resident elephant population but primarily serves elephants owned by local communities. Its fluctuating monthly workload appears influenced by external factors, particularly elephant relocation and care-seeking behavior. During the COVID-19 pandemic (mid-2021 to early 2022), many elephants were returned from tourism camps to home provinces, especially Surin, due to nationwide lockdowns and travel restrictions [7,14]. This shift coincided with a temporary surge in admissions, reaching up to 30 cases per month over a 10-month span in 2022. As restrictions lifted and tourism resumed, DLD's workload declined accordingly [14]. Furthermore, Surin province hosts another facility—The Elephant Kingdom Hospital—operated by the Zoological Park Organization. Located in Baan Ta Klang, Thailand's largest elephant village [7], it offers more convenient access than DLD, which is situated approximately 100 km away. These overlapping services and geographic considerations likely influence owner preferences and hospital utilization patterns. Such findings emphasize the need for deeper examination of admission dynamics across facilities, including seasonal movement, informal care practices, and regional access barriers. Collectively, the contrasting model performances reflect fundamental differences in the patient demographics (e.g., more geriatric chronic cases at NEI), regional elephant management (tourism vs. private ownership), and healthcare-seeking behaviors.

### 4.2. Seasonal disease trends

Seasonal analysis showed a clear pattern in hospital admissions at NEI, but no distinct seasonality at DLD. These differences likely reflect regional variation in elephant use and management. In northern Thailand, elephants are heavily involved in the tourism-related activities such as feeding, bathing, and riding [1,44]. In contrast, elephants in Surin and nearby provinces are predominantly privately owned, allowing owners greater flexibility to engage them in cultural ceremonies, breeding programs, and live social media broadcasts [7]. These divergent activity patterns may influence both the timing and nature of health issues requiring hospital care.

Although trend and seasonal analyses were based on total active cases, variation in seasonal disease patterns was evident between hospitals, despite similar dominant health issues. In both NEI and DLD, integumentary and gastrointestinal disorders were most common. However, at NEI, integumentary cases peaked in December-January, aligning with the peak tourism season in northern Thailand [44,45]. This pattern likely reflects increased elephant workload during high visitor periods, consistent with previous findings linking skin lesion to intensive tourist interactions [46]. Gastrointestinal cases showed a bimodal seasonal peak: hospitalizations rose in April–May (late dry season), likely due to poor-quality

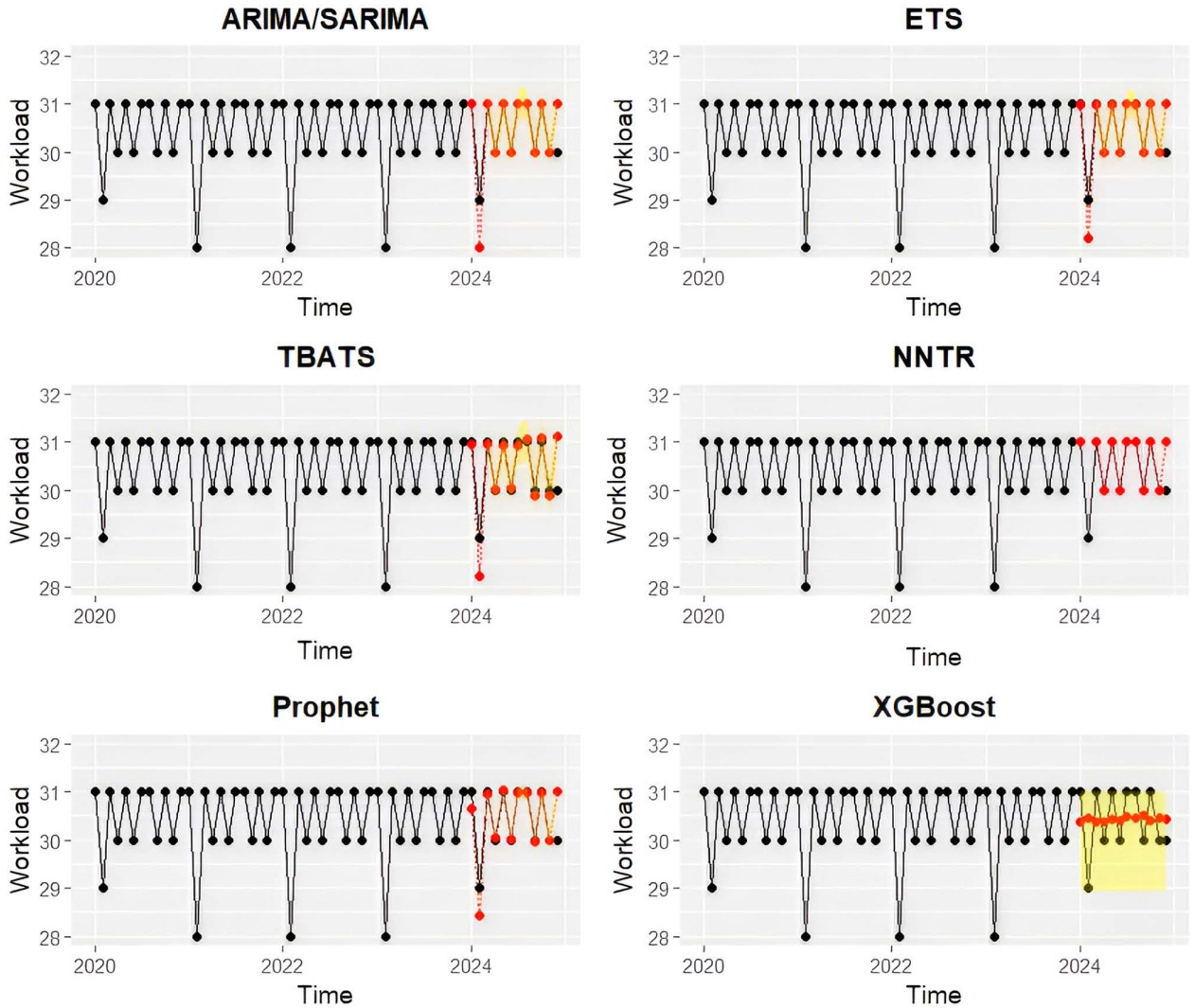

**Fig 3. Actual active monthly workload admissions at NEI elephant hospital (black dots and lines) from January 2020 to December 2024, along with forecasted values for January to December 2024 (red dots and lines) generated from six times series models trained on the training dataset (January 2020 to December 2023).** Auto Regressive Integrated Moving Average model and Seasonal ARIMA (ARIMA/SARIMA), Exponential Smoothing model (ETS), Trigonometric, Box-Cox transformation, ARMA errors, Trend, and Seasonal components model (TBATS), Neural Network Time Series Regression (NNTR), Prophet, and Extreme Gradient Boosting (XGBoost). The yellow area represented 95% confidence intervals.

forage increasing colic risk [12,13], and again in July–August (mid-rainy season), potentially linked to elevated soil inges-tion during foraging [47,48]. EEHV-HD cases occurred year-round but peaked in February and July, partially aligned with national trends reported by Yun et al. [10], where highest incidence occurred during Thailand's climatic transition from hot-humid to rainy conditions.

Unlike NEI, the DLD elephant hospital did not display clear seasonal patterns in admissions, suggesting either a more uniform distribution of health issues or differences in management and reporting practices. The absence of seasonality may not reflect true epidemiological uniformity, but rather underreporting. In Surin, many mahouts manage non-severe conditions at home, rather than seeking hospital care—often due to distance, cost, or convenience—mirroring trends observed in Myanmar [49,50]. As a result, hospital records likely underrepresent milder cases and may obscure seasonal

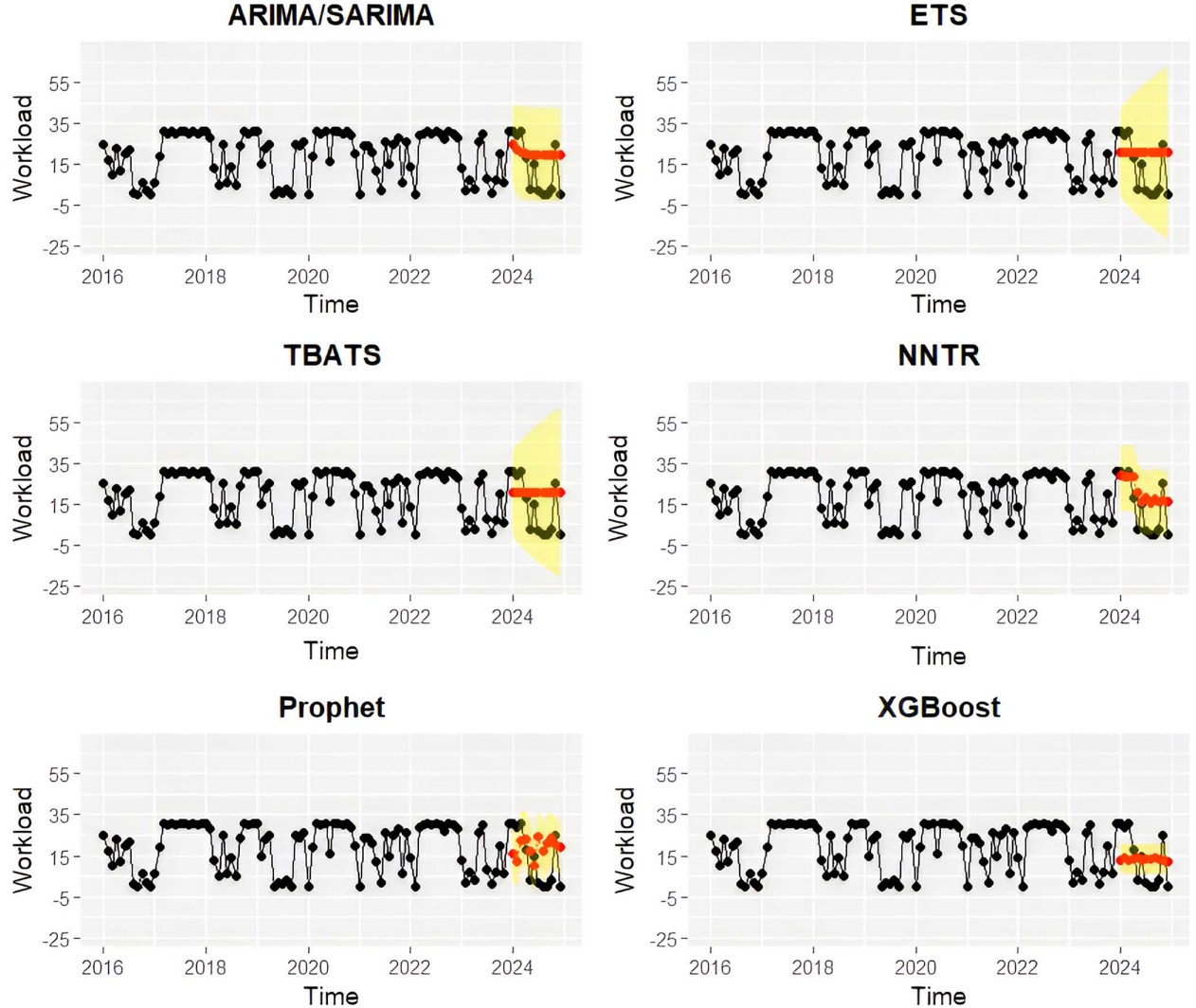

**Fig 4. Actual active monthly workload admissions at DLD elephant hospital (black dots and lines) from January 2016 to December 2024, along with forecasted values for January to December 2024 (red dots and lines) generated from six times series models trained with the training dataset (January 2016 to December 2023).** Auto Regressive Integrated Moving Average model and Seasonal ARIMA (ARIMA/SARIMA), Exponential Smoothing model (ETS), Trigonometric, Box-Cox transformation, ARMA errors, Trend, and Seasonal components model (TBATS), Neural Network Time Series Regression (NNTR), Prophet, and Extreme Gradient Boosting (XGBoost). The yellow area represented 95% confidence intervals.

fluctuations. Additionally, Surin has lower tourism traffic than northern Thailand, reducing the influence of seasonal visitor peaks on elephant workload and injury rates [44,51,52]. At DLD, gastrointestinal disorders remained the most common issues, with modest increases observed in April (N = 14), October (N = 10), and November (N = 9). The April rise corresponds with the dry season, when elephants are often tethered in dry fields and may experience inconsistent feeding. The post-harvest peak in October-November likely reflects overconsumption of rice straw in paddies, increasing colic risk [53]. No strong seasonal trends were found for integumentary or EEHV-HD cases. Wound-related conditions may be managed privately or through mobile veterinary clinics [5,51], while EEHV-HD cases may go unrecorded due to early mortality or empirical treatment with antivirals at home [7,54]. These findings highlight the limitation of relying solely on hospital-based

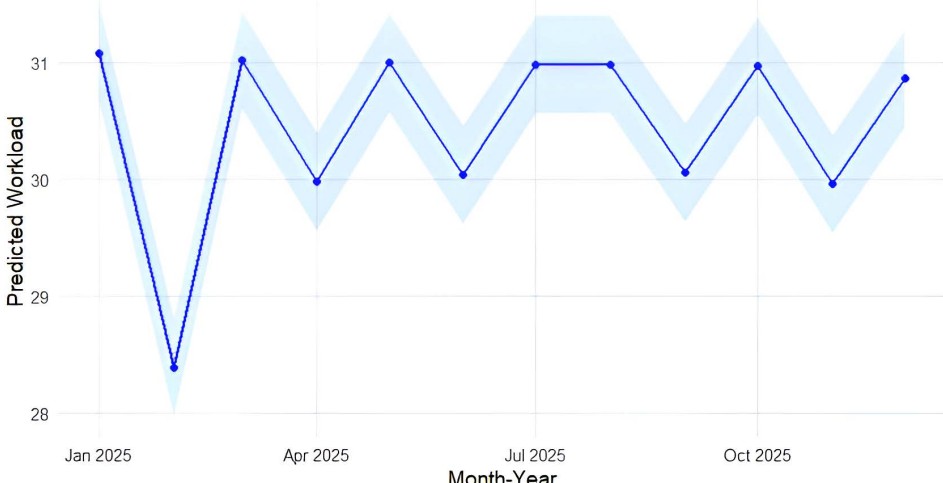

**Fig 5. Forecast of monthly admission workload (blue dots and lines) at the National Elephant Institute elephant hospital (NEI) from January to December 2025 using the Exponential Smoothing model (ETS).** The light blue area represented the 95% confidence intervals.

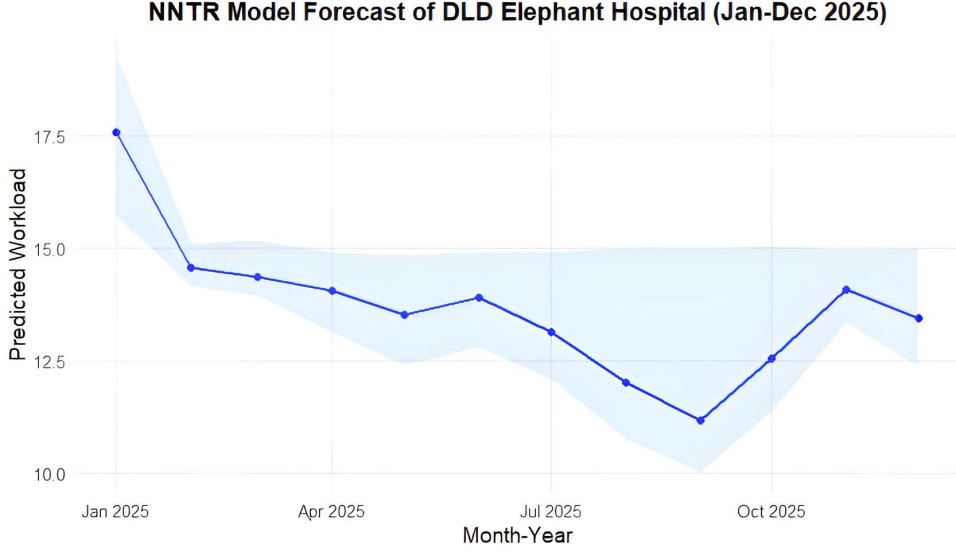

**Fig 6. Forecast of monthly admission workload (blue dots and lines) at Department of Livestock Development elephant hospital (DLD) from January to December 2025 using the Neural Network Time Series Regression model (NNTR).** The light blue area represented the 95% confidence intervals generated using bootstrapping methods.

surveillance and underscore the need for broader, long-term datasets to accurately assess seasonal trends in elephant health.

### 4.3. Model interpretability

Model comparison showed that ETS performed best for NEI, while NNTR outperformed other models at DLD. ETS was well-suited to NEI's stable, seasonal admission trends [20,55], whereas NNTR effectively captured the irregular and

non-linear fluctuations observed at DLD [15,19,56]. Other models, including ARIMA/SARIMA, performed less favorably, likely due to their dependence on strong seasonality assumptions [18,35], which were absent at DLD and only moderately present in NEI's five-year dataset. TBATS, though designed for complex seasonal patterns [20,38], did not enhance predictive accuracy beyond ETS, possibly because NEI's seasonality was already sufficiently captured. Prophet, which incorporates trend and holiday effects [29,41], showed only moderate accuracy and struggled with short-term variations. XGBoost, a machine-learning approach, failed to outperform classical models, likely due to the small dataset size and absence of external predictors such as climate variables or management interventions [20,57,58]. These findings highlight the importance of aligning model selection with the underlying structure of each dataset, rather than defaulting to complex algorithms without justification.

### 4.4.  Forecast implications and systemic gaps

Despite differences in historical trends, the forecasted workload patterns at NEI and DLD diverged notably. NEI's projected caseload remained stable throughout 2025, suggesting consistent demand for long-term care, likely reflecting its role in managing resident elephants and complex chronic cases. A slight dip in February aligned with observed seasonal lulls, potentially linked to reduced tourist activity during the warmer months [44,51]. In contrast, DLD's workload showed a gradual decline, with the lowest caseload forecasted in September. This trend does not correspond to observed seasonal peaks, implying reduced hospital utilization rather than a shift in disease burden. One possible explanation is the seasonal migration of working elephants, as mahouts often relocate for labor or return home during rice harvesting in October-November. Additionally, some cases may be treated in the field, bypassing hospital records. These dynamics may contribute to underreporting during certain periods. The downward trend in DLD forecasts highlights the need to explore whether this reflects a true decline in caseload or changes in care-seeking behavior, accessibility, or data completeness. These gaps highlight the importance of triangulating hospital data with field-level surveillance to refine forecasting accuracy. Importantly, even in resource-limited veterinary contexts, forecasting holds practical utility. Projections can inform medication procurement schedules, guide staffing needs during peak periods, and support facility preparedness. The ability to shift from reactive to anticipatory logistics is especially valuable in endangered species care, where treatment delays can be fatal and resource shortages can jeopardize animal welfare.

### 4.5.  Limitations and future directions

Several limitations should be acknowledged. First, only admission-based hospital data were analyzed. Cases managed via mobile clinics and during routine health checks (e.g., blood collection, deworming) were excluded, potentially underestimating true healthcare demand. Second, data availability differed between sites (five years for NEI, nine for DLD), limiting model comparability and trend detection. This discrepancy stem from reliance on paper-based records which introduced risks of missing or incomplete data and affecting historical accuracy. Third, the sample size remains modest due to the rarity of elephant hospitalization, which may constrain model generalizability. Despite these limitations, meaningful insights were obtained, reinforcing the value of digitalization and centralized record systems for wildlife health monitoring to improve data quality and support reliable forecasting.

Future research should integrate external variables such as local management practices, tourist seasonality, and climatic conditions to improve model robustness. Including mobile clinic data and expanding coverage across additional elephant hospitals would enhance workload estimation and epidemiological surveillance. However, model performance must be reassessed with each expanded dataset. Establishing a centralized national elephant database would support more generalizable forecasting frameworks, contribute to proactive healthcare planning, and align with conservation goals under One Health and welfare-centered approaches.

## 5. Conclusion

This study demonstrated that time-series forecasting, while traditionally applied in high-volume human health settings, can be effectively adapted to wildlife hospitals with low patient volume but high per-case resource demands. By evaluating six forecasting models across two major elephant hospitals in Thailand (NEI and DLD), we identified hospital-specific caseload dynamics and optimal model performance tailored to each context. The stable, seasonal pattern caseload at NEI was best captured by the ETS model, reflecting its role in chronic, long-term care. In contrast, the irregular admissions at DLD, more accurately modeled by NNTR, underscored the influence of external factors such as tourism, owner-driven care-seeking, and local access barrier. Our findings emphasize the critical importance of selecting forecasting models that align with the temporal structure and operational realities of each hospital. Crucially, this study provides the first demonstration that meaningful, actionable forecasts can be generated from sparse, real-world datasets in endangered species care. These predictions offer operational value by informing staffing plans, guiding medical supply procurement, and strengthening emergency readiness. As such, this work serves as a first template for data-driven planning in wildlife hospitals, an area previously unsupported by forecasting science. To improve forecasting robustness and generalizability, future research should incorporate broader variables such as mobile clinic data, climatic conditions, seasonal tourism patterns, and elephant movement dynamics. Expanding digital health infrastructure and establishing a centralized national database would further enhance real-time monitoring, resource sharing, and long-term planning. As the veterinary care of captive wildlife becomes increasingly complex, the integration of predictive analytics into hospital systems offers a scalable, conservation-aligned strategy to improve both animal welfare and healthcare system resilience.

## Supporting information

**S1 Fig. Autocorrelation function (ACF) plots of monthly active caseloads at NEI (left) and DLD (right) elephant hospitals.** NEI showed sustained positive autocorrelation across short lags, consistent with underlying seasonal or cyclical workload patterns. In contrast, DLD displayed weak or no autocorrelation, indicating stochastic or irregular admission dynamics.
(PDF)

**S1 Table. Statistical comparison between NEI and DLD hospital datasets.**
(PDF)

## Acknowledgments

We would like to thank Dr. Warangkana Langkapin and Dr. Janjeera Kaew-Ek for their kind support.

## Author contributions

**Conceptualization:** Worapong Kosaruk, Veerasak Punyapornwithaya.

**Data curation:** Worapong Kosaruk.

**Formal analysis:** Worapong Kosaruk.

**Funding acquisition:** Chatchote Thitaram.

**Investigation:** Worapong Kosaruk, Pichamon Ueangpaibool, Nawapa Hirannithithamrong.

**Methodology:** Worapong Kosaruk, Veerasak Punyapornwithaya, Pichamon Ueangpaibool, Nawapa Hirannithithamrong.

**Project administration:** Veerasak Punyapornwithaya, Chatchote Thitaram.

**Software:** Worapong Kosaruk, Veerasak Punyapornwithaya.

**Supervision:** Veerasak Punyapornwithaya, Chatchote Thitaram.

**Validation:** Veerasak Punyapornwithaya, Chatchote Thitaram.

**Visualization:** Worapong Kosaruk.

**Writing – original draft:** Worapong Kosaruk.

**Writing – review & editing:** Worapong Kosaruk, Veerasak Punyapornwithaya, Chatchote Thitaram, Pichamon Ueangpaibool, Nawapa Hirannithithamrong.

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
