## [Decision Letter · Decision Letter 0]

29 Jul 2025

Dear Dr. Kosaruk,

Thank you for submitting your manuscript to PLOS ONE. After careful consideration, we feel that it has merit but does not fully meet PLOS ONE’s publication criteria as it currently stands. Therefore, we invite you to submit a revised version of the manuscript that addresses the points raised during the review process.

We look forward to receiving your revised manuscript.

Kind regards,

Michael Döllinger, Ph.D.

Academic Editor

PLOS ONE

Journal Requirements:

Additional Editor Comments:

Although the work is of interest, as both reviewers and the editor think, the manuscript requires substantial revisions in structure, clarity, and justification to reach its full potential, see comments. If the authors cannot improve that in the revised version, the manusrcipt still has to be rejected in the next round.

Reviewers' comments:

Reviewer's Responses to Questions

**Comments to the Author**

1. Is the manuscript technically sound, and do the data support the conclusions?

Reviewer #1: Partly

Reviewer #2: Partly

2. Has the statistical analysis been performed appropriately and rigorously?

Reviewer #1: No

Reviewer #2: Yes

3. Have the authors made all data underlying the findings in their manuscript fully available?

Reviewer #1: Yes

Reviewer #2: Yes

4. Is the manuscript presented in an intelligible fashion and written in standard English?

Reviewer #1: No

Reviewer #2: No

Reviewer #1: The provided article implementing several time-series estimation methods to forecast the workload of elephant hospitals. Although the author's justification of the novelty of the suggested idea and domain sounds reasonable, the article lacks the reasonable justification and explanation about the experimental results. Consequently, the provided article is not formatted to be an appropriate scientific article which is the reason why I suggest to reject the given paper.

To be specific major concerns provided below should be considered

Q1. Is there significant difference exist in dataset of NEI and DLD? If there exists, the author should provide and justify such difference. If there exists no significant difference, the author should explain why all six model demonstrate significantly different performance in two distinct dataset in Table 1.

Q2. Although the author suggests prediction results in NEI and DLD dataset, I cannot find the main claim of the author that shows the main contribution of the presented work. If it is natural to have different prediction results in different dataset, I declare that two distinct dataset is too small set to show the novelty of the work. If the author targeted to make the general forecasting methods, the generalization of the method in distinct dataset is absence. Therefore, I cannot find the main claim of the given paper to have the contribution in the domain.

Q3. Please write the paper to be clear and sound reasonable.

Reviewer #2: The article addresses an important and unique problem—adapting time series forecasting approaches used for human infectious diseases to veterinary medicine, namely, for elephants. The creation and analysis of a digital dataset from two hospitals in Thailand, specifically for elephants, is a valuable contribution. However, there are several critical areas that need attention:

1. Justification of the Study:

While studying elephant health is undoubtedly valuable, particularly given their endangered status (as listed by the IUCN), the rationale for forecasting illness rates among such a small population needs further clarification. Is there a known or emerging infectious disease affecting elephants in Thailand that motivates this study? Given the limited number of existing elephants, the number of potentially sick individuals may be too low to justify complex forecasting. Additionally, the normal operational capacity of the hospitals should be described to understand the practical implications of a potential increase in cases.

2. Ambiguity in Analysis:

The choice of time windows for the two hospitals is inconsistent—five years for one and nine for the other. A consistent time frame would allow for better comparison across models and datasets. The rationale behind this discrepancy should be clearly explained, or the data range should be standardized.

3. Paper Structure:

The manuscript would benefit from a major structural revision. Sectioning is insufficient, and content is often presented in inappropriate sections. For instance, the methodological pipeline is briefly introduced in the Data section (lines 128–132), which is not ideal. A clearer separation between data description, methodology, results, and discussion would greatly improve readability and comprehension.

4. Clarity of Writing:

The article would benefit from careful editing to improve clarity and precision. For example, it states in the Abstract that “six time-series forecasting models” were used, but these are not listed at the time of mention. Furthermore, the statement that “Exponential Smoothing (ETS) demonstrated the highest accuracy for NEI” in the abstract lacks context—highest compared to which models? Evaluation metrics such as MAE, RMSE, and MASE are used without being defined or briefly explained in the manuscript, which may confuse readers unfamiliar with these terms.

Conclusion:

Overall, the topic and dataset are promising and relevant. However, the manuscript requires substantial revisions in structure, clarity, and justification to reach its full potential.

**Do you want your identity to be public for this peer review?** For information about this choice, including consent withdrawal, please see our Privacy Policy

Reviewer #1: No

Reviewer #2: No

---

## [Author Response · Author response to Decision Letter 1]

15 Aug 2025

Thank you very much for the thoughtful and constructive feedback from the Academic Editor and both reviewers. We truly appreciate the opportunity to revise our manuscript.

We have carefully addressed all comments in a detailed, point-by-point Response to Reviewers, which is submitted as a separate file. In this revised version, we have made substantial improvements to the structure, clarity, and justification of the manuscript. Key changes include clearer sectioning of the Methods and Discussion, improved explanation of model performance differences, and more precise use of terminology. We have also clarified our rationale, defined evaluation metrics, and expanded on data availability.

We hope the revised manuscript now meets the journal’s expectations, and we sincerely thank the editorial team for their guidance throughout this process.

---

## [Decision Letter · Decision Letter 1]

13 Nov 2025

Time-series analysis for forecasting monthly workload at two elephant hospitals in Thailand

PONE-D-25-18082R1

Dear Dr. Kosaruk,

We’re pleased to inform you that your manuscript has been judged scientifically suitable for publication and will be formally accepted for publication once it meets all outstanding technical requirements.

Kind regards,

Michael Döllinger, Ph.D.

Academic Editor

PLOS ONE

Additional Editor Comments (optional):

Pleass answer to the final comments of the questions - then the manuscript can be accpted for publication

Reviewers' comments:

Reviewer's Responses to Questions

**Comments to the Author**

Reviewer #2: All comments have been addressed

2. Is the manuscript technically sound, and do the data support the conclusions?

Reviewer #2: Yes

3. Has the statistical analysis been performed appropriately and rigorously?

Reviewer #2: Yes

4. Have the authors made all data underlying the findings in their manuscript fully available?

Reviewer #2: No

5. Is the manuscript presented in an intelligible fashion and written in standard English?

Reviewer #2: Yes

Reviewer #2: The authors have done an excellent job in thoroughly addressing my earlier concerns — thank you for the thoughtful revisions. I believe the manuscript could become even stronger with the inclusion of the following references for added comprehensiveness:

• https://doi.org/10.54254/2754-1169/2024.19252

• https://doi.org/10.1371/journal.pone.0282624

• https://doi.org/10.1016/j.procs.2021.01.036

• https://doi.org/10.1109/AISP61396.2024.10475218

• https://doi.org/10.1007/s11356-023-25148-9

**Do you want your identity to be public for this peer review?** For information about this choice, including consent withdrawal, please see our Privacy Policy

Reviewer #2: No

---

## [Editor Report · Acceptance letter]

PONE-D-25-18082R1

PLOS One

Dear Dr. Kosaruk,

I'm pleased to inform you that your manuscript has been deemed suitable for publication in PLOS One. Congratulations! Your manuscript is now being handed over to our production team.

Kind regards,

on behalf of

Dr. Michael Döllinger

Academic Editor

PLOS One